# NR2F2 is required in the embryonic testis for fetal Leydig cell development

Aitana Perea-Gomez*, Natividad Bellido Carreras, Magali Dhellemmes, Furong Tang, Coralie Le Gallo, Marie-Christine Chaboissier*

Université Côte d'Azur, CNRS, INSERM, iBV, Nice, France

## eLife Assessment

This **important** study, which has been improved further upon revision, reveals a critical role of the transcription factor NR2F2 in mouse fetal Leydig cell (FLC) differentiation. With elegantly carried out experiments, the authors provide **compelling** evidence that NR2F2 helps to initiate the differentiation of certain interstitial cells into FLC until these cells mature into functional secretory cells that produce androgen and insulin-like peptide 3 (INSL3). The particular importance of the work comes from the fact that NR2F2 affects FLCs without altering paracrine signals known to be involved in FLC differentiation. The work will be of interest to colleagues studying reproductive development in mammals including humans or the biological functions of the nuclear receptor family.

*For correspondence:
Aitana.PEREA-GOMEZ@univ-cotedazur.fr (AP-G);
marie-christine.chaboissier@univ-cotedazur.fr (M-ChristineC)

Competing interest: The authors declare that no competing interests exist.

## Abstract

Male genital development in XY mammalian fetuses is triggered by the action of hormones, including testosterone, secreted by the developing testes. Defects in this process are a cause for differences in sex development (DSD), one of the most common congenital abnormalities in humans. Fetal Leydig cells (FLCs) play a central role in the synthesis of masculinizing hormones in the developing testes. Yet, the genetic cascade controlling their differentiation is poorly understood. Here, we investigate the role of the orphan nuclear receptor NR2F2 (COUP-TFII) in FLC development. We report that NR2F2 is expressed in interstitial progenitor cells of the mouse embryonic testes and is downregulated upon their differentiation into FLC. By using two mouse models for conditional mutation of *Nr2f2* in the developing testes, we demonstrate that NR2F2 is required for testis morphogenesis and FLC development. NR2F2 acts in interstitial progenitors to regulate the initiation and progression of FLC differentiation. These results establish NR2F2 as an essential regulator of FLC development and steroid hormone synthesis in the mouse fetal testis and provide an entry point in understanding the etiology of 46,XY DSD associated with pathogenic NR2F2 variants.

## Introduction

Sexual development in mammals is conditioned by the gonadal sex established during fetal life. In XY embryos, the initially undifferentiated gonads develop into testes that can produce testosterone which stimulates the differentiation of male internal and external genitalia (including epididymis, vas deferens, seminal vesicles, scrotum, and penis) leading to the masculinization of the fetus. In contrast, fetal ovaries do not produce testosterone, and female internal and external genitalia (oviducts, uterus, vagina, and vulva) develop in XX individuals. Abnormalities in the masculinization process are mainly associated with defects in androgen synthesis or signaling (*Reyes et al., 2023*). Although this molecular cascade is well defined, the differentiation of androgen-producing cells in the embryo is only partially understood.

Androgen production in the developing testis relies mainly on fetal Leydig cells (FLCs), which express all enzymes required for the biosynthesis of androstenedione from cholesterol (*Miyabayashi*

*et al., 2017*; *Inoue et al., 2016*; *Shima et al., 2013*; *Ademi et al., 2022*). The final step of conversion of androstenedione into testosterone, catalyzed by HSD17B3, takes place in a different cell type, the fetal Sertoli cells (*Shima et al., 2013*; *O'Shaughnessy et al., 2000*). In addition to their role in androgen synthesis, FLCs also produce INSL3, a hormone required for testis descent (*Nef and Parada, 1999*; *Zimmermann et al., 1999*). Defects in testis descent, regulated also by androgens, result in cryptorchidism, a condition which impacts fertility and constitutes a risk factor for testicular cancer (*Hutson et al., 2015*). While a fraction of FLCs persist after birth, others de-differentiate or involute, and adult testosterone production is ensured by a distinct population of steroidogenic cells, the adult Leydig cells (ALCs), which differentiate at puberty (*Shima and Morohashi, 2017*; *Shima, 2019*).

FLCs differentiate from embryonic day 12.5 (E12.5) in mice and increase in number during fetal life through the recruitment and differentiation of *Wnt5a* positive proliferative progenitors located in the interstitial space of the testes (*Ademi et al., 2022*; *Shima and Morohashi, 2017*; *Shima, 2019*; *Rotgers et al., 2018*). Interstitial progenitors also give rise to the contractile peritubular myoid cells (PTM) that will surround the future seminiferous tubules (*Ademi et al., 2022*). Despite being the most abundant cell population in the fetal testis (*Ademi et al., 2022*; *Mayère et al., 2022*), little is known about the genetic control of the proliferation, specification, and differentiation of the interstitial steroidogenic progenitor cells.

Lineage tracing and single-cell transcriptomic analyses have revealed that interstitial progenitors have a dual origin. The majority are derived from the coelomic epithelium of the undifferentiated gonad, which harbors early bipotential progenitors able to differentiate along the supporting (the future Sertoli cells) or the steroidogenic lineage (*Ademi et al., 2022*; *Karl and Capel, 1998*; *Stévant et al., 2018*; *Stévant et al., 2019*). In addition, Nestin positive cells migrate from the adjacent mesonephros into the gonad and differentiate along the steroidogenic lineage from E13.5 to give rise to up to a third of the FLC population by the end of gestation (*Ademi et al., 2022*; *DeFalco et al., 2011*; *Kumar and DeFalco, 2018*).

Positive and negative paracrine signals drive FLC differentiation by upregulating the transcription factors NR5A1, GATA4, and GATA6, which in turn regulate the expression of genes related to cholesterol metabolism and steroidogenesis (*Rotgers et al., 2018*; *Wen et al., 2016*; *Morohashi et al., 2020*; *Baba et al., 2018*; *Baba et al., 2014*; *Shima et al., 2018*; *Padua et al., 2015*; *Bielinska et al., 2007*; *Viger et al., 2022*). Desert Hedgehog (DHH) is secreted by Sertoli cells and acts on the interstitial progenitors expressing the Hedgehog receptor Patched1 (PTCH1) and the Hedgehog effectors GLI1, -2, and -3 to trigger FLC and PTM differentiation (*Yao et al., 2002*; *Barsoum and Yao, 2011*; *Barsoum et al., 2009*; *Park et al., 2000*; *Kothandapani et al., 2020*; *Pierucci-Alves et al., 2001*; *Clark et al., 2000*). In addition, FLC development requires the activation of signaling pathways downstream of PDGFRA in the steroidogenic progenitors (*Brennan et al., 2003*; *Inoue et al., 2022*; *Schmahl et al., 2008*). On the other hand, ligands present in vascular and perivascular cells activate the NOTCH2 receptor and the expression of the effectors HES1 and HEYL in interstitial progenitors to maintain their undifferentiated state and inhibit FLC differentiation (*Kumar and DeFalco, 2018*; *Tang et al., 2008*; *Liu et al., 2016*; *Defalco et al., 2013*). In addition to the paracrine signals from adjacent cell populations, FLC differentiation is also regulated by the cell-autonomous action of the transcription factors ARX, TCF21, PBX1, MAF, and MAFB expressed in the interstitial steroidogenic progenitors, although their precise roles remain elusive (*Ademi et al., 2022*; *Miyabayashi et al., 2013*; *Li et al., 2021*; *Cui et al., 2004*; *Schnabel et al., 2003*).

Nuclear receptor subfamily 2 group F member 2 (NR2F2, also known as COUP-TFII) is abundantly expressed in interstitial progenitors of the fetal and adult testis in rodents and humans (*Inoue et al., 2016*; *Ademi et al., 2022*; *Mendoza-Villarroel et al., 2014b*; *Kilcoyne et al., 2014*; *van den Driesche et al., 2012*; *Lottrup et al., 2014*; *Qin et al., 2008*; *Taelman et al., 2024*). NR2F2 activates or represses transcription depending on the cellular context by directly binding to DNA responsive elements or by interacting with other transcription factors. NR2F2 regulates cell differentiation during organogenesis, adult tissue homeostasis, and tumorigenesis (*Polvani et al., 2019*). NR2F2 function is essential for cardiac and vascular development so that *Nr2f2* mutation in mouse leads to embryonic lethality at mid-gestation (*Pereira et al., 1999*). The study of mouse conditional mutants has shown that NR2F2 is essential for ALC differentiation in the postnatal testis before puberty (*Qin et al., 2008*). However, the function of NR2F2 in the developing testis during fetal life has not been addressed, and its role in the interstitial progenitors that give rise to the FLC lineage remains unknown. It was initially

proposed that NR2F2 could act as a negative regulator of steroidogenesis at fetal stages based on the inverse correlation between NR2F2 expression and steroidogenesis genes and testosterone levels in mouse and rat fetal testes treated with endocrine disruptors (*van den Driesche et al., 2012*). More recently, rare variants in NR2F2 have been associated with cryptorchidism, hypospadias, and defective penile growth in human patients (*Zidoune et al., 2022*; *Ganapathi et al., 2023*; *Wankanit et al., 2024*). These phenotypes can be attributed to defective testosterone and INSL3 production during gestation (*Hutson et al., 2015*; *Amato et al., 2022*), suggesting a positive role for NR2F2 in promoting FLC differentiation and/or function in the fetal testis.

In this study, we show that NR2F2 is expressed in interstitial progenitors of coelomic and mesonephric origin of the mouse fetal testes and is downregulated upon FLC differentiation. By using two Cre lines that drive *Nr2f2* deletion in mouse embryonic gonads, we show that NR2F2 is required for fetal mouse testis morphogenesis and for FLC differentiation. Absence of *Nr2f2* does not impair paracrine signals known to regulate FLC differentiation nor the proliferation or survival of the steroidogenic progenitor population. Our findings reveal that NR2F2 promotes the initiation of FLC differentiation as well as FLC maturation. Taken together, these results establish NR2F2 as an essential factor that positively regulates the development of steroidogenic cells in the mouse fetal testis.

## Results

### NR2F2 is expressed in steroidogenic progenitors of the developing testis

We analyzed the spatiotemporal distribution of NR2F2 in the developing testis by immunofluorescence. At E11.5 (18–21 tail somites [ts]), NR2F2 was detected in both the coelomic epithelium and the mesonephric mesenchyme adjacent to the gonads, two tissues that contribute to the population of interstitial steroidogenic progenitors (*Ademi et al., 2022*; *Karl and Capel, 1998*; *Kumar and DeFalco, 2018*; *Figure 1A and C*). The majority of mesenchymal cells in the gonad expressed the transcription factor RUNX1 (RUNX1+) (*Nicol et al., 2019*), indicating that most of the gonadal somatic cells at this stage belong to the supporting lineage (*Figure 1A and D*). Nevertheless, NR2F2+ RUNX1- cells were observed, revealing that interstitial progenitors were already present at this stage (*Figure 1A*). NR2F2+ cells were either GATA4+ or GATA4-, suggesting that interstitial progenitors of coelomic and mesonephric origins, respectively, were both present (*Figure 1A–C*).

In E12.5 testes, NR2F2+ cells were detected in the coelomic epithelium and in the interstitial space outside the developing testis cords (*Figure 1E–H*, *Figure 1—figure supplement 1A–D*). NR2F2+ cells co-expressed the steroidogenic progenitor marker ARX (*Ademi et al., 2022*; *Miyabayashi et al., 2013*) and were actively proliferating (*Figure 1E–H*, *Figure 1—figure supplement 1E–H*). At E14.5, NR2F2 expression was maintained in interstitial cells co-expressing PDGFRA (*Brennan et al., 2003*), as well as in the PTM lining the testis cords and in the cells beneath the surface of the testis that will contribute to the future tunica albuginea (*Figure 1I and J*, *Figure 1—figure supplement 1M–P*). NR2F2 expression was also found in NESTIN-expressing perivascular cells that correspond to the mesonephros-derived steroidogenic progenitors (*Kumar and DeFalco, 2018*; *Figure 1K and L*). In contrast, NR2F2 protein was either absent or detected at very low levels in FLC marked by the expression of the steroidogenic enzyme HSD3B (*Figure 1M–P*, *Figure 1—figure supplement 1I–L*).

Together, our results show that NR2F2 is expressed in the coelomic epithelium and the mesonephros, as well as in the interstitial progenitors derived from both sources, and is downregulated upon FLC differentiation in the developing testis.

### NR2F2 is required for fetal testicular morphogenesis and FLC development

In order to investigate the function of NR2F2 in the developing mouse testis, we used an *Nr2f2^flox* conditional allele (*Hutson et al., 2015*), and a knock-in *Wt1^CreERT2* line, in which tamoxifen-inducible CreERT2 is produced by *Wt1*-expressing cells (*Shima and Morohashi, 2017*; *Figure 2A*). *Wt1* is expressed from E9.5 in the coelomic epithelium of the gonadal ridge and in the adjacent mesonephros (*Armstrong et al., 1993*), and *Wt1^CreERT2*-mediated recombination can be induced in all somatic gonadal cells upon tamoxifen administration at E9.5 and E10.5 (*Manuylov et al., 2008*; *Figure 2—figure supplement 1A*).

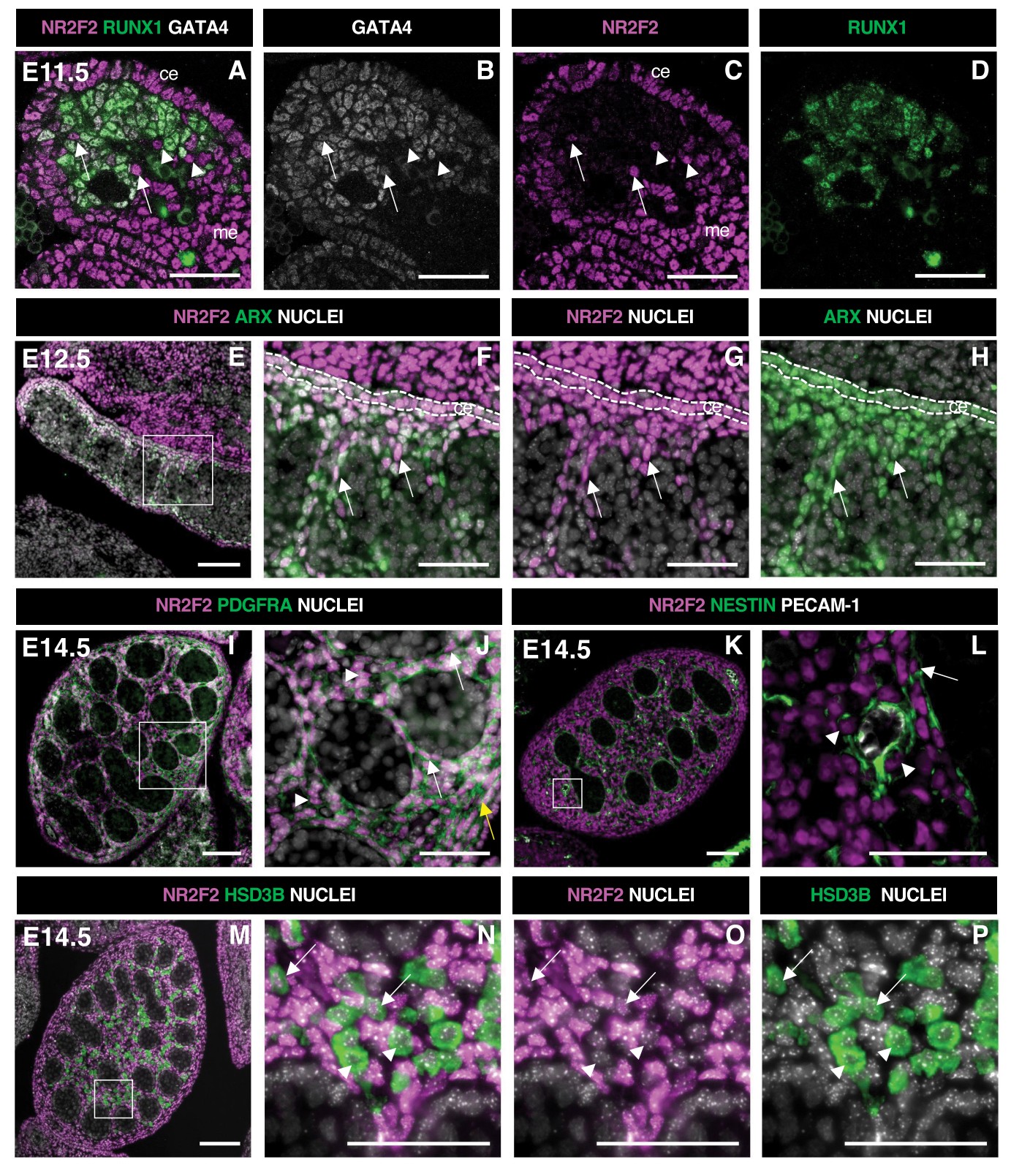

**Figure 1.** NR2F2 is expressed in steroidogenic progenitors of the fetal testis. (**A–D**) Immunodetection of NR2F2, RUNX1, and GATA4 on embryonic day 11.5 (E11.5) (18 tail somites) XY gonad. NR2F2 is detected in coelomic epithelium (ce), mesonephros (me), and RUNX1 negative cells that are either GATA4 positive (arrows in A–C) or GATA4 negative (arrowheads in A–C). (**E–H**) Immunodetection of NR2F2 and ARX on E12.5 XY gonad. NR2F2 is co-expressed with ARX in the coelomic epithelium (ce, dotted lines in F–H) and in interstitial cells between the testis cords (arrows in F–H). (**I,J**)

*Figure 1 continued on next page*

*Figure 1 continued*

Immunodetection of NR2F2 and PDGFRA on E14.5 XY gonad. NR2F2 is detected in PDGFRA positive cells, including interstitial progenitors (arrowheads in J), peritubular myoid cell surrounding testis cords (arrows in J), and cells of the future tunica albuginea (yellow arrow in J). (**K,L**) Immunodetection of NR2F2, NESTIN, and PECAM-1 on E14.5 XY gonad. NR2F2 is detected in NESTIN positive interstitial progenitors, including perivascular cells (arrowheads in L) and peritubular myoid cells (arrow in L). (**M–P**) Immunodetection of NR2F2 and HSD3B on E14.5 XY gonad. NR2F2 is absent from the majority of HSD3B positive fetal Leydig cells (arrowheads in N–P) and is only detected at low levels in a few HSD3B positive elongated cells (arrows in N–P). Data are representative of triplicate biological replicates. Scale bar = 50 µm in A–D, F–H, J, L, and N–P. Scale bar = 100 µm in E, I, K, M.

The online version of this article includes the following figure supplement(s) for figure 1:

**Figure supplement 1.** NR2F2 is expressed in steroidogenic progenitors of the fetal testis.

NR2F2 is co-expressed with WT1 in the gonadal coelomic epithelium, in the mesonephros, and in interstitial cells (*Yu et al., 2012*; *Figure 2—figure supplement 1C–F*). Tamoxifen treatment at E9.5 and E10.5 triggered an efficient NR2F2 deletion in gonadal and mesonephric tissues of $Wt1^{CreERT2}$; $Nr2f2^{flox/flox}$ embryos analyzed at E12.5 and E14.5 (*Figure 2A–E*, *Figure 2—figure supplement 1B*). NR2F2 expression was completely absent in gonadal interstitial cells, including NESTIN+ cells, demonstrating that all interstitial steroidogenic progenitors were targeted in $Wt1^{CreERT2}$; $Nr2f2^{flox/flox}$ embryos (*Figure 2D and E*). Morphological examination of the urogenital system of E16.5 $Wt1^{CreERT2}$; $Nr2f2^{flox/flox}$ embryos revealed hypoplastic undescended testes, as well as hypoplastic kidneys (*Figure 2H and I*), indicating that NR2F2 function is required for testicular and kidney development.

Sertoli cells expressing SOX9 (*Figure 2B and C*) and interstitial progenitors marked by ARX (*Figure 2F and G*) were present in $Wt1^{CreERT2}$; $Nr2f2^{flox/flox}$ embryos at E12.5. However, differentiated steroidogenic FLC (marked by HSD3B expression) were almost completely absent in the mutant gonads (*Figure 2J, K, and N*). This phenotype was not due to a delay in the initiation of FLC differentiation, as the number of HSD3B positive cells remained strongly reduced in $Wt1^{CreERT2}$; $Nr2f2^{flox/flox}$ mutant gonads at E14.5 (60% reduction, *Figure 2L–N*). In addition, the transcripts of *Cyp11a1* and *Cyp17a1*, two genes encoding steroidogenic enzymes expressed in FLC, and of *Insl3* were strongly reduced (*Figure 2O*), providing a possible explanation for the observed undescended testes phenotype (*Figure 2—figure supplement 1G and H*). The mutant gonads showed fewer testis cords that were enlarged and exhibited irregular shapes at E14.5 (*Figure 2E*, *Figure 2—figure supplement 1J and M*). Moreover, the expression of ACTA2 was strongly reduced both in the periphery of the gonad and in cells lining the testis cords, indicating that tunica cell and PTM development were impaired in $Wt1^{CreERT2}$; $Nr2f2^{flox/flox}$ mutants (*Figure 2—figure supplement 1G and H*). Together, these results demonstrate that NR2F2 function is not essential for the initial specification of the interstitial and supporting cells of the testis but is required for testicular morphogenesis and for FLC development.

## Sertoli cell development is impaired in $Wt1^{CreERT2}$; $Nr2f2^{flox/flox}$ testes

FLC differentiation relies on signals produced by Sertoli cells such as DHH and PDGFA acting on interstitial cells (*Rotgers et al., 2018*; *Wen et al., 2016*). The number of Sertoli cells expressing SOX9 and the levels of *Sox9* transcripts were not altered in $Wt1^{CreERT2}$; $Nr2f2^{flox/flox}$ mutants at E14.5 (*Figure 2N and O*). However, the expression of *Dhh*, *Pdgfa*, and *Amh*, another marker of differentiated Sertoli cells, was reduced in $Wt1^{CreERT2}$; $Nr2f2^{flox/flox}$ mutants compared to $Nr2f2^{flox/+}$ control gonads (*Figure 2—figure supplement 1K*), indicating that Sertoli cell differentiation is abnormal in $Wt1^{CreERT2}$; $Nr2f2^{flox/flox}$ gonads. WT1 is required for Sertoli cell development (*Gao et al., 2006*), and previous work suggested that *Wt1* heterozygosity in $Wt1^{CreERT2/+}$ gonads results in delayed testicular development (*Carré et al., 2018*). To discriminate between the effects of *Nr2f2* loss of function and *Wt1* heterozygosity in the phenotype of $Wt1^{CreERT2}$; $Nr2f2^{flox/flox}$ mutants, we analyzed gene expression in $Wt1^{CreERT2/+}$ testes compared to wild-type littermates. We found that the expression of the FLC markers *Cyp11a1* and *Cyp17a1* was not significantly different in $Wt1^{CreERT2/+}$ testes compared to wild-type littermates (*Figure 2—figure supplement 1L*). In contrast, transcript levels of *Dhh*, *Pdgfa*, and *Amh* were significantly reduced (*Figure 2—figure supplement 1L*). While these results indicate that impaired FLC development in $Wt1^{CreERT2}$; $Nr2f2^{flox/flox}$ mutants is associated with the loss of *Nr2f2* function, the potential contribution of Sertoli cell defects caused by *Wt1* heterozygosity to the FLC phenotype cannot be ruled out in this model.

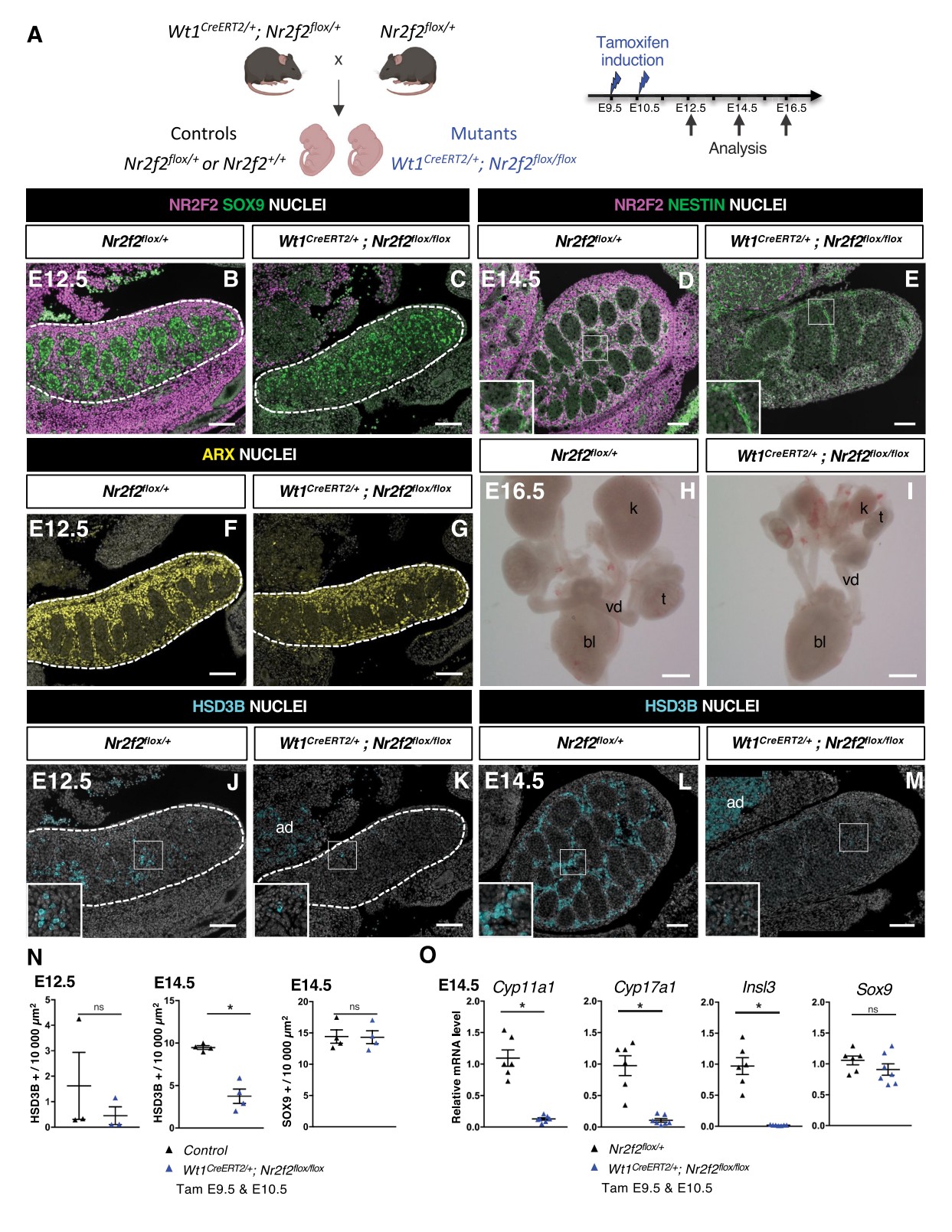

**Figure 2.** NR2F2 deletion by *Wt1^CreERT2^* impairs Sertoli cell differentiation and fetal Leydig cell (FLC) development. (**A**) Generation of *Wt1^CreERT2^*; *Nr2f2^flox/flox^* mutants. Tamoxifen was administered at embryonic day 9.5 (E9.5) and E10.5, and embryos were recovered at E12.5, E14.5, and E16.5. (**B,C**) Immunodetection of NR2F2 and SOX9 on E12.5 *Nr2f2^flox/+^* and *Wt1^CreERT2^*; *Nr2f2^flox/flox^* testes (outlined by dotted lines) after tamoxifen treatment at E9.5 and E10.5. NR2F2 is efficiently deleted in the gonad and mesonephros. (**D,E**) Immunodetection of NR2F2 and NESTIN on E14.5 *Nr2f2^flox/+^* and

*Figure 2 continued on next page*

*Figure 2 continued*

*Wt1^{CreERT2}; Nr2f2^{flox/flox}* testes. NR2F2 is efficiently deleted in NESTIN1 positive cells. (**F,G**) Immunodetection of ARX on E12.5 *Nr2f2^{flox/+}* and *Wt1^{CreERT2};* *Nr2f2^{flox/flox}* testes (outlined by dotted lines). Interstitial cells are generated in *Wt1^{CreERT2}; Nr2f2^{flox/flox}* mutants. (**H,I**) Macroscopic view of the urogenital tract of XY E16.5 *Nr2f2^{flox/+}* and *Wt1^{CreERT2}; Nr2f2^{flox/flox}* testes dissected after tamoxifen treatment at E9.5 and E10.5. Testes (**t**) and kidneys (**k**) are hypoplastic in *Wt1^{CreERT2}; Nr2f2^{flox/flox}* mutants. bl: bladder. vd: vas deferens. (**J,K**) Immunodetection of HSD3B on E12.5 *Nr2f2^{flox/+}* and *Wt1^{CreERT2}; Nr2f2^{flox/flox}* testes (outlined by dotted lines). ad: adrenal. (**L,M**) Immunodetection of HSD3B on E14.5 *Nr2f2^{flox/+}* and *Wt1^{CreERT2}; Nr2f2^{flox/flox}* testes. ad: adrenal. (**N**) Quantification of the number of HSD3B positive cells per surface unit and of the number of SOX9 positive cells per surface unit in control (wild-type or *Nr2f2^{flox/+}*) and *Wt1^{CreERT2}; Nr2f2^{flox/flox}* testes. Each triangle represents the mean number of HSD3B or SOX9 positive cells per surface unit of one individual measured on at least two sections per gonad. Data are shown as means ± SEM. Statistical significance was assessed by Mann-Whitney U two-tailed test. * indicates p-value≤0.05; ns indicates p-value>0.05. (**O**) Quantification of *Cyp11a1, Cyp17a1, Insl3,* and *Sox9* transcripts in *Nr2f2^{flox/+}* and *Wt1^{CreERT2}; Nr2f2^{flox/flox}* testes treated with tamoxifen at E9.5 and E10.5 and dissected at E14.5 after normalization to *Sdha* and *Tbp* by RT-qPCR. Data are shown as means ± SEM. Statistical significance was assessed by Mann-Whitney U two-tailed test. * indicates p-value≤0.05; ns indicates p-value>0.05. Immunodetection data are representative of triplicate biological replicates. Scale bar = 100 μm in B–G, J–M. Scale bar = 500 μm in H, I.

The online version of this article includes the following source data and figure supplement(s) for figure 2:

**Source data 1.** Source data for cell counts and RT-qPCR data in *Figure 2*.

**Figure supplement 1.** NR2F2 deletion by *Wt1^{CreERT2}* impairs Sertoli cell differentiation and fetal Leydig cell (FLC) development.

**Figure supplement 1—source data 1.** Source data for testis cord measurements and RT-qPCR data in *Figure 2—figure supplement 1*.

## NR2F2 is required in the steroidogenic lineage for FLC development

In order to elucidate the specific function of NR2F2 in the steroidogenic lineage for FLC development, we used the transgenic *Nr5a1-Cre* line which drives robust recombination in somatic gonadal cells from E11.5, after the supporting and steroidogenic lineages have been specified (*Figure 3A*, *Figure 3—figure supplement 1A and B*; *Bingham et al., 2006*; *Manuylov et al., 2011*).

At E11.5, NR5A1 is co-expressed with NR2F2 in the gonadal coelomic epithelium and in interstitial cells, but is absent from the mesonephros and mesonephros-derived cells (*Stévant et al., 2018*; *Kumar and DeFalco, 2018*; *Figure 3—figure supplement 1D–G*). *Nr5a1-Cre; Nr2f2^{flox/flox}* mutants show efficient *Nr2f2* deletion in gonadal interstitial cells at E12.5 and E14.5 (*Figure 3B–E*, *Figure 3—figure supplement 1C*). Consistent with previous reports on the activity of *Nr5a1-Cre* (*Manuylov et al., 2011*), NR2F2 was still detected in the coelomic epithelium layer and in interstitial cells just beneath it, particularly in the anterior part of the gonad of *Nr5a1-Cre; Nr2f2^{flox/flox}* mutants (*Figure 3B–E*). NESTIN+ NR2F2+ cells were still present, confirming that steroidogenic progenitors of mesonephric origin were not targeted by *Nr5a1-Cre* (*Figure 3D and E*, *Figure 3—figure supplement 1H and I*; *Kumar and DeFalco, 2018*). Together, these results demonstrate that in *Nr5a1-Cre; Nr2f2^{flox/flox}* mutants, NR2F2 is deleted after E11.5 in interstitial cells derived from the coelomic epithelium, except in the outermost layer of the testis.

*Nr2f2* deletion by *Nr5a1-Cre* did not affect the initial formation of the supporting and steroidogenic lineages as evidenced by SOX9 (*Figure 3B, C, and H*, *Figure 3—figure supplement 1N and O*) and ARX expression (*Figure 3F and G*), nor the differentiation of Sertoli cells as shown by normal expression levels of *Dhh, Pdgfa,* and *Amh* (*Figure 3H*, *Figure 3—figure supplement 1N*). In addition, ACTA2 was detected in PTM and was only slightly reduced in the tunica albuginea of the posterior region in the mutant testes (*Figure 3—figure supplement 1J and K*). In contrast, the FLC population marked by HSD3B or CYP11A1 was decreased (40% reduction, *Figure 3I–M*, *Figure 3—figure supplement 1L and M*), and the expression of the FLC markers *Cyp11a1, Cyp17a1,* and *Insl3* was strongly downregulated in *Nr5a1-Cre; Nr2f2^{flox/flox}* mutant testes (*Figure 3N*). We conclude that *Nr2f2* deletion after E11.5 by *Nr5a1-Cre* leads to FLC reduction without Sertoli cell defects, suggesting that NR2F2 is required cell-autonomously in the interstitial cells for FLC development.

In agreement with reduced *Insl3* expression, the testes of *Nr5a1-Cre; Nr2f2^{flox/flox}* embryos were undescended and exhibited an abnormal abdominal position at postnatal day (P) 3 (*Figure 3O and P*). In addition, the anogenital distance, a readout for androgen levels (*Schwartz et al., 2019*), was reduced in P0 and P1 *Nr5a1-Cre; Nr2f2^{flox/flox}* males when compared to male controls (*Figure 3—figure supplement 1P*), consistent with the reduced expression of steroidogenic enzymes in the mutant testes. We conclude that defective FLC development in *Nr2f2* mutants results in cryptorchidism and impaired masculinization of the external genitalia.

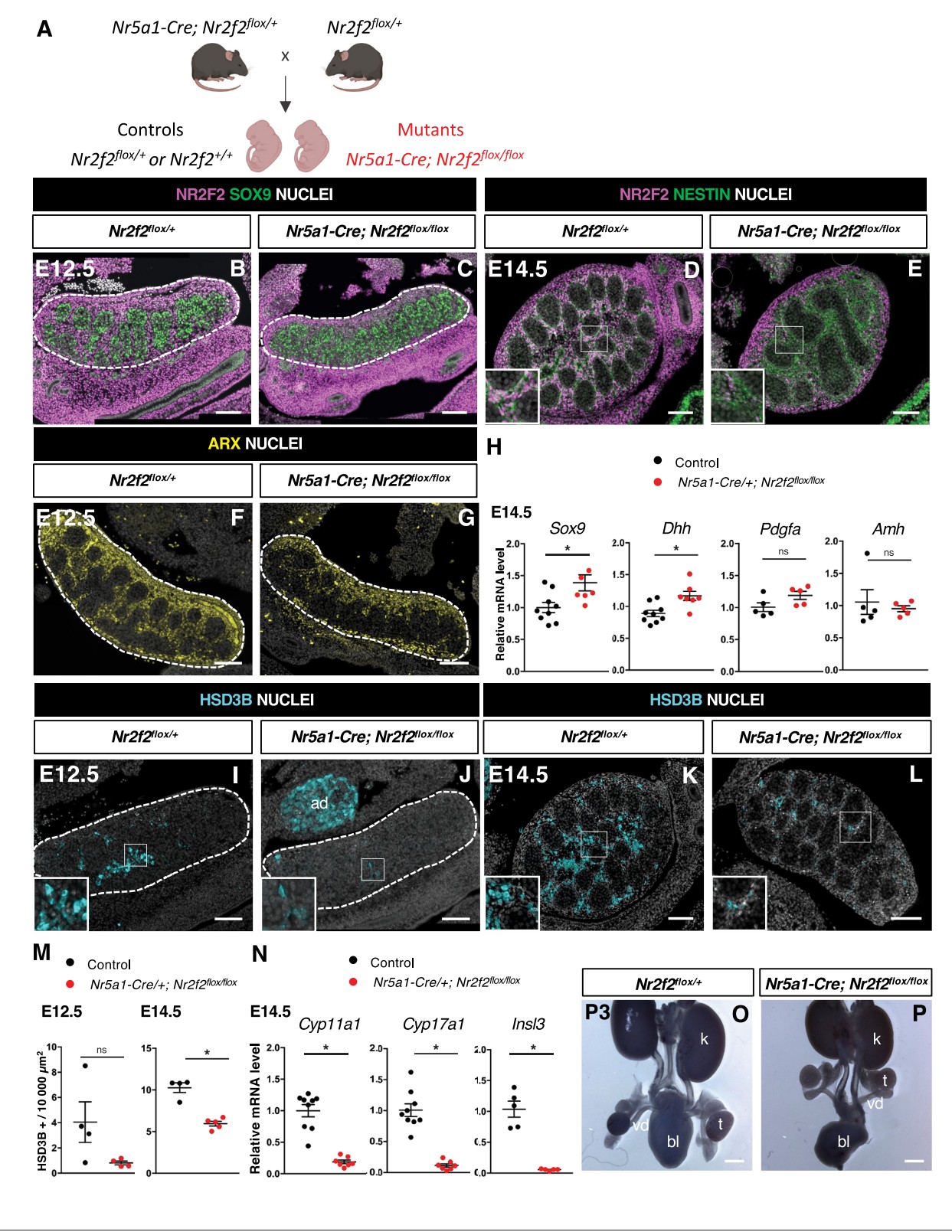

**Figure 3.** NR2F2 deletion by Nr5a1-Cre impairs fetal Leydig cell (FLC) development. (**A**) Generation of *Nr5a1-Cre; Nr2f2^flox/flox* mutants. (**B,C**) Immunodetection of NR2F2 and SOX9 on embryonic day 12.5 (E12.5) *Nr2f2^flox/+* and *Nr5a1-Cre; Nr2f2^flox/flox* testes (outlined by dotted lines). NR2F2 is deleted in interstitial cells but is still present in the outermost layer of the testis. (**D,E**) Immunodetection of NR2F2 and NESTIN on E14.5 *Nr2f2^flox/+* and *Nr5a1-Cre; Nr2f2^flox/flox* testes. NR2F2 is still detected in NESTIN1 positive cells. (**F,G**) Immunodetection of ARX on E12.5 *Nr2f2^flox/+* and *Nr5a1-Cre;*

*Figure 3 continued on next page*

Figure 3 continued

*Nr2f2flox/flox* testes (outlined by dotted lines). Interstitial cells are generated in *Nr5a1-Cre; Nr2f2flox/flox* mutants. (H) Quantification of *Sox9, Dhh, Pdgfa,* and *Amh* transcripts after normalization to *Sdha* and *Tbp* in control (wild-type or *Nr2f2flox/+*) and *Nr5a1-Cre; Nr2f2flox/flox* by RT-qPCR at E14.5. Data are shown as means ± SEM. Statistical significance was assessed by Mann-Whitney U two-tailed test. * indicates p-value≤0.05; ns indicates p-value>0.05. (I,J) Immunodetection of HSD3B on E12.5 *Nr2f2flox/+* and *Nr5a1-Cre; Nr2f2flox/flox* testes (outlined by dotted lines). ad: adrenal. (K,L) Immunodetection of HSD3B on E14.5 *Nr2f2flox/+* and *Nr5a1-Cre; Nr2f2flox/flox* testes. (M) Quantification of the number of HSD3B positive cells per surface unit in control (wild-type or *Nr2f2flox/+*) and *Nr5a1-Cre; Nr2f2flox/flox* testes at E12.5 and E14.5. Each circle represents the mean number of HSD3B positive cells per surface unit of one individual measured on at least two sections per gonad. Data are shown as means ± SEM. Statistical significance was assessed by Mann-Whitney U two-tailed test. * indicates p-value≤0.05; ns indicates p-value>0.05. (N) Quantification of *Cyp11a1, Cyp17a1,* and *Insl3* after normalization to *Sdha* and *Tbp* in control (wild-type or *Nr2f2flox/+*) and *Nr5a1-Cre; Nr2f2flox/flox* by RT-qPCR at E14.5. Data are shown as means ± SEM. Statistical significance was assessed by Mann-Whitney U two-tailed test. * indicates p-value≤0.05; ns indicates p-value>0.05. (O,P) Macroscopic view of the urogenital tract of XY P3 *Nr2f2flox/+* and *Nr5a1-Cre; Nr2f2flox/flox* mutants. The testes are in abdominal position in *Nr5a1-Cre; Nr2f2flox/flox* mutants. k: kidney, t: testis, bl: bladder. vd: vas deferens. Immunodetection data are representative of triplicate biological replicates. Scale bar = 100 μm in B–G, I–L. Scale bar = 500 μm in O, P.

The online version of this article includes the following source data and figure supplement(s) for figure 3:

**Source data 1.** Source data for cell counts and RT-qPCR data in *Figure 3*.

**Figure supplement 1.** NR2F2 deletion by Nr5a1-Cre impairs fetal Leydig cell (FLC) development.

**Figure supplement 1—source data 1.** Source data for cell counts, anogenital distance measurments and RT-qPCR data in *Figure 3—figure supplement 1*.

## NR2F2 is required for the initiation of FLC differentiation

FLCs differentiate from proliferating interstitial progenitors that progressively lose their mitotic ability, downregulate the transcription factors ARX and NR2F2, upregulate the master regulator of steroidogenesis NR5A1, and activate the expression of steroidogenesis-related genes (*Inoue et al., 2016*; *Ademi et al., 2022*; *Miyabayashi et al., 2013*). We wanted to determine which of these steps of FLC differentiation are NR2F2 dependent.

The transcription factor ARX is required in the pool of proliferating interstitial progenitors for FLC development (*Miyabayashi et al., 2013*). We first examined whether NR2F2 regulates the survival, proliferation, or identity of the ARX+ cell population. We found that the expression of *Arx* mRNA was not modified in *Nr5a1-Cre; Nr2f2flox/flox* mutants (*Figure 4—figure supplement 1A*). The total number of gonadal cells, the percentage of ARX+ cells among the total number of gonadal cells, and the fraction of proliferating cells among the ARX+ population were similar to controls at E12.5 and E14.5 (*Figure 4A, B, and E*, *Figure 4—figure supplement 1B–E*). In addition, we did not find evidence of increased cell death in *Nr5a1-Cre; Nr2f2flox/flox* mutants (*Figure 4—figure supplement 1F–H*). Together, these results indicate that ARX+ steroidogenic progenitor cells are present and proliferate at normal rates in *Nr5a1-Cre; Nr2f2flox/flox* mutants.

As the ARX+ steroidogenic progenitors adopt an FLC identity, NR2F2 is progressively lost, and the nuclear receptor NR5A1 is strongly upregulated (*Inoue et al., 2016*; *Ademi et al., 2022*; *van den Driesche et al., 2012*). NR5A1 directs FLC differentiation by regulating the expression of genes associated with cholesterol metabolism and steroidogenesis (*Morohashi et al., 2020*; *Shima et al., 2018*; *Jeyasuria et al., 2004*; *Buaas et al., 2012*). In *Nr5a1-Cre; Nr2f2flox/flox* mutants, the cells expressing high levels of NR5A1+ in the interstitial compartment were reduced compared to controls (*Figure 4C–E*, *Figure 4—figure supplement 1I and J*). This observation indicates that NR2F2 function is required for NR5A1 upregulation at the initial step of steroidogenic cell differentiation.

## DHH, PDGFRA, and NOTCH pathways are not impaired in *Nr5a1-Cre; Nr2f2flox/flox* mutants

We next investigated the impact of *Nr5a1-Cre; Nr2f2 flox/flox* mutation on the activation of the signaling pathways involved in FLC differentiation. DHH secreted by Sertoli cells binds to its receptor PTCH1 expressed in interstitial progenitors and activates its target genes, including *Gli1*, to promote FLC differentiation (*Yao et al., 2002*; *Barsoum and Yao, 2011*). The expression levels of *Dhh* and of its target *Gli1*, a readout of Hedgehog pathway activation, were not modified in *Nr5a1-Cre; Nr2f2flox/flox* mutants (*Figure 3H*, *Figure 4F*, *Figure 3—figure supplement 1N*). *Gli1* transcripts exhibited a similar distribution in interstitial cells of control and *Nr5a1-Cre; Nr2f2flox/flox* mutants at E14.5 (*Figure 4G and H*). PDGFA produced by Sertoli cells binds to the PDGFRA receptor expressed in interstitial progenitors to activate downstream signaling required for FLC development (*Brennan et al., 2003*; *Schmahl*

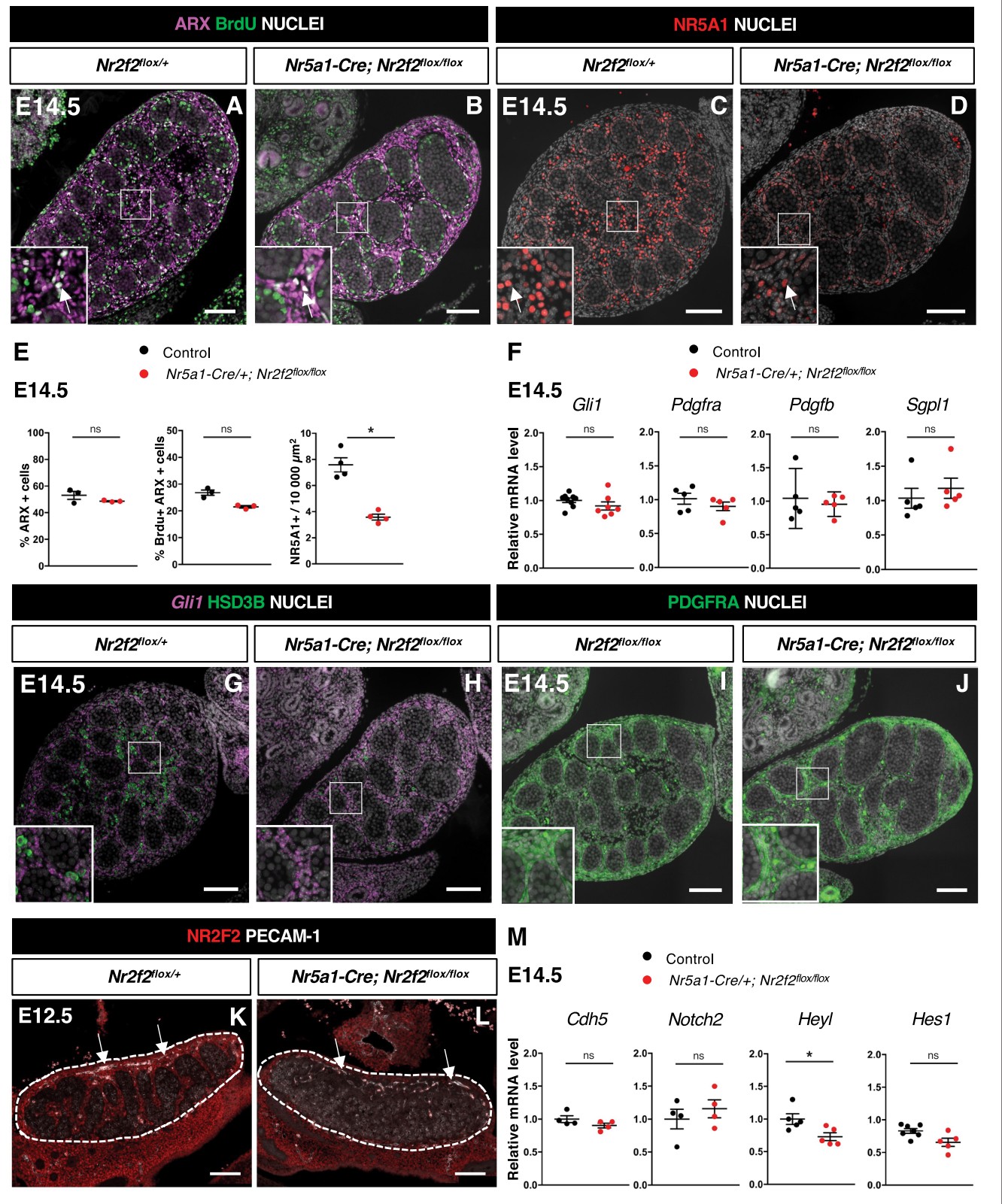

**Figure 4.** NR2F2 function is required for the initiation of fetal Leydig cell (FLC) differentiation. (**A,B**) Immunodetection of ARX and BrdU (arrows) on embryonic day 14.5 (E14.5) XY *Nr2f2flox/+* and *Nr5a1-Cre; Nr2f2flox/flox* testes. (**C,D**) Immunodetection of NR5A1 on E14.5 XY *Nr2f2flox/+* and *Nr5a1-Cre; Nr2f2flox/flox* testes. Arrows indicate strong expression of NR5A1 positive FLC. (**E**) Quantification of the percentage of ARX positive cells (number of ARX positive nuclei relative to the total number of nuclei labeled by DAPI), of the percentage of ARX positive cells labeled by BrdU (number of nuclei positive

*Figure 4 continued*

for ARX and BrdU relative to the number of ARX positive nuclei) and of the number of NR5A1 positive cells per surface unit in E14.5 XY control (wild-type or *Nr2f2^flox/+*) and *Nr5a1-Cre; Nr2f2^flox/flox* testes. Each circle represents the mean percentage of ARX+ or ARX+/BrdU+ or NR5A1+ cells per surface unit of one individual measured on at least two sections per gonad. Data are shown as means ± SEM. Statistical significance was assessed by Mann-Whitney U two-tailed test. * indicates p-value≤0.05; ns indicates p-value>0.05. (**F**) Quantification of *Gli1*, *Pdgfra*, *Pdgfb*, and *Sgpl1* transcripts after normalization to *Sdha* and *Tbp* in control (wild-type or *Nr2f2^flox/+*) and *Nr5a1-Cre; Nr2f2^flox/flox* by RT-qPCR at E14.5. Statistical significance was assessed by Mann-Whitney U two-tailed test. * indicates p-value≤0.05; ns indicates p-value>0.05. (**G,H**) In situ hybridization detection of *Gli1* transcripts and immunodetection of HSD3B protein on E14.5 XY *Nr2f2^flox/+* and *Nr5a1-Cre; Nr2f2^flox/flox* testes. (**I,J**) Immunodetection of PDGFRA on E14.5 XY *Nr2f2^flox/flox* and *Nr5a1-Cre; Nr2f2^flox/flox* testes. (**K,L**) Immunodetection of PECAM-1 and NR2F2 on E12.5 XY *Nr2f2^flox/+* and *Nr5a1-Cre; Nr2f2^flox/flox* testes. PECAM-1 is expressed in germ cells and in endothelial cells (white arrows). (**M**) Quantification of Cdh5, *Notch2, Heyl*, and *Hes1* transcripts after normalization to *Sdha* and *Tbp* in control (wild-type or *Nr2f2^flox/+*) and *Nr5a1-Cre; Nr2f2^flox/flox* by RT-qPCR at E14.5. Statistical significance was assessed by Mann-Whitney U two-tailed test. * indicates p-value≤0.05; ns indicates p-value>0.05. Immunodetection data are representative of triplicate biological replicates. Scale bar = 100 μm.

The online version of this article includes the following source data and figure supplement(s) for figure 4:

**Source data 1.** Source data for cell counts and RT-qPCR data in *Figure 4*.

**Figure supplement 1.** NR2F2 function is required for the initiation of fetal Leydig cell (FLC) differentiation.

**Figure supplement 1—source data 1.** Source data for cell counts and RT-qPCR data in *Figure 4—figure supplement 1*.

*et al., 2008*). The expression levels of *Pdgfa* and *Pdgfb*, coding for two PDGFRA ligands expressed in developing testes (*Brennan et al., 2003*), were not altered in *Nr5a1-Cre; Nr2f2^flox/flox* mutants (*Figures 3H and 4F*). Transcript levels of *Pdgfra* were unchanged in *Nr5a1-Cre; Nr2f2^flox/flox* mutants (*Figure 4F*), and PDGFRA protein was detected at the plasma membrane of interstitial cells in both control and *Nr5a1-Cre; Nr2f2^flox/flox* mutant testes at E14.5 (*Figure 4I and J*). In addition, the expression of *Sgpl1*, a PDGFRA signaling target involved in steroidogenic differentiation (*Schmahl et al., 2008*), was not reduced in *Nr5a1-Cre; Nr2f2^flox/flox* mutants (*Figure 4F*). These observations indicate that the activities of Hedgehog and PDGFRA signaling, two pathways that positively regulate FLC differentiation, are not impaired in *Nr5a1-Cre; Nr2f2^flox/flox* mutants.

In addition to the positive signals, FLC differentiation is also negatively modulated by NOTCH signaling triggered by ligands expressed in vascular and perivascular cells. An increase in testicular endothelial cells results in a reduction in FLC numbers (*Kumar and DeFalco, 2018*; *Tang et al., 2008*; *Liu et al., 2016*; *Defalco et al., 2013*). The distribution and abundance of endothelial cells marked by PECAM-1 was not altered in *Nr5a1-Cre; Nr2f2^flox/flox* mutants (*Figure 4K and L*). In addition, *Cdh5* transcript levels (a readout of the abundance of endothelial cells) were similar in controls and *Nr5a1-Cre; Nr2f2^flox/flox* mutants (*Figure 4M*). These observations indicate that the reduction in FLCs in *Nr5a1-Cre; Nr2f2^flox/flox* mutants was not associated with an increased population of endothelial cells. The receptor NOTCH2 expressed in interstitial cells is involved in restricting FLC differentiation (*Kumar and DeFalco, 2018*; *Tang et al., 2008*; *Liu et al., 2016*; *Defalco et al., 2013*). *Notch2* mRNA levels were not altered in *Nr5a1-Cre; Nr2f2^flox/flox* mutants (*Figure 4M*). NOTCH signaling activates the expression of its target genes (that also act as effectors of the pathway), including *Heyl*, specifically expressed in interstitial cells and strongly upregulated upon vascular depletion (*Kumar and DeFalco, 2018*) and *Hes1*, expressed in interstitial cells and involved in restricting FLC differentiation (*Tang et al., 2008*; *Liu et al., 2016*). Transcripts of the NOTCH pathway targets *Hes1* and *Heyl* were detected at similar levels in controls and *Nr5a1-Cre; Nr2f2^flox/flox* mutants (*Figure 4M*), indicating that the observed decrease in FLC numbers cannot be attributed to an increase in NOTCH signaling.

Together, these results indicate that NR2F2 deficiency does not impact the activity of the signaling pathways known to induce or repress FLC formation, suggesting that NR2F2 acts downstream or in parallel to these pathways to regulate FLC differentiation from interstitial progenitors.

## NR2F2 is required for FLC maturation

A small fraction of HSD3B positive cells formed in the absence of NR2F2 function, even in the case of *Wt1^CreERT2; Nr2f2^flox/flox* mutant gonads that exhibit widespread NR2F2 deletion in steroidogenic progenitors of coelomic and mesonephric origin. One possibility is that these cells are ALCs that have prematurely differentiated in *Nr2f2* mutant fetal testes. *Bhmt* (*Betaine-homocysteine methyltransferase*) is specifically expressed in ALC and absent in FLC (*Sararols et al., 2021*). *Bhmt* transcripts were not detected in control or *Nr2f2* mutant testes by RT-qPCR at E14.5, while they were present in

adult testes (*Figure 5—figure supplement 1A*), indicating that the steroidogenic cells of the mutants do not have adult characteristics.

To evaluate the steroidogenic capacities of the remaining FLC in *Nr2f2* mutant gonads, we analyzed the transcript levels of *Cyp11a1* and *Cyp17a1* normalized to the FLC number as quantified by HSD3B immunofluorescence (*Figure 2N*, *Figure 3M*; *Kothandapani et al., 2020*). Normalized data showed reduced steroidogenic gene expression in *Nr5a1-Cre; Nr2f2^{flox/flox}* and in *Wt1^{CreERT2}; Nr2f2^{flox/flox}* mutant testes (*Figure 5A and B*), in agreement with the reduced HSD3B expression levels detected by immunofluorescence in *Nr2f2* mutant testes (*Figures 2J–M and 3I–L*). These observations suggest that FLC formed in the mutant testes have reduced steroidogenic function.

FLC differentiation is accompanied by a change in cell shape from spindle-shaped progenitors to round-shaped FLC and an increase in cytoplasmic volume (*van den Driesche et al., 2012*; *Haider, 2004*). HSD3B positive FLC present in *Nr5a1-Cre; Nr2f2^{flox/flox}* and in *Wt1^{CreERT2}; Nr2f2^{flox/flox}* mutants were smaller and more elongated than those in control littermates (*Figure 5C–H*, *Figure 5—figure supplement 1*). These cellular characteristics have been associated with immature FLC at the initial stages of FLC differentiation before the formation of large and round testosterone-producing FLC (*van den Driesche et al., 2012*; *Haider, 2004*).

Taken together, our results show that NR2F2 is required in interstitial steroidogenic progenitor cells for initiation of FLC differentiation but also for the subsequent step of FLC maturation leading to robust steroid production (*Figure 5I*).

## Discussion

NR2F2 protein is expressed in interstitial cells of coelomic epithelium and mesonephric origin and is absent or detected at very low levels in FLC as soon as they are formed at E12.5. Our observations are consistent with NR2F2 positive cells being progenitors for FLC (*Inoue et al., 2016*; *Ademi et al., 2022*; *Estermann et al., 2025*) and indicate that NR2F2 is quickly downregulated upon differentiation of the fetal steroidogenic lineage, when the cells begin to synthesize steroid hormones.

We used two Cre lines to address the role of NR2F2 in the mouse fetal testis. *Nr2f2* mutants exhibit reduced FLC numbers together with a strong reduction in *Insl3* and steroidogenic gene expression, resulting in cryptorchidism and reduced anogenital distance. Importantly, the reduction of *Insl3* expression is sufficient to impair testis descent even when a fraction of FLC is still present. Our findings identify NR2F2 as an essential regulator of FLC and male reproductive system development in the mouse. This conclusion is confirmed by an independent *Nr2f2* conditional mutation resulting in a similar phenotype of FLC hypoplasia, cryptorchidism, and hypospadias (*Estermann et al., 2025*).

*Nr2f2* deletion using *Nr5a1-Cre* does not impair the survival or proliferation of the interstitial steroidogenic progenitor population, yet the FLC population is reduced in *Nr5a1-Cre; Nr2f2^{flox/flox}* mutant testes. We found that the expression of the master regulator of steroidogenic differentiation of NR5A1 is not upregulated in *Nr2f2* mutants, which is sufficient to account for the decrease in FLC numbers (*Morohashi et al., 2020*; *Baba et al., 2018*; *Baba et al., 2014*; *Shima et al., 2018*). ChIP-seq analysis of whole E14.5 testes has identified NR2F2 binding peaks in the regulatory region of *Nr5a1* (*Estermann et al., 2025*). In addition, NR2F2 regulates gene expression by directly interacting with NR5A1 in MA-10 cells, an in vitro model for immature ALCs (*Mendoza-Villarroel et al., 2014b*; *Mendoza-Villarroel et al., 2014a*; *Di-Luoffo et al., 2022*). It would be interesting to explore whether an interaction between NR2F2 and NR5A1 contributes to the strong upregulation of *Nr5a1* itself in differentiating FLCs and/or whether NR2F2 cooperates with other factors for the regulation of *Nr5a1*.

The activity of the main paracrine pathways regulating FLC differentiation was not significantly altered in *Nr5a1-Cre; Nr2f2^{flox/flox}* mutant testes, as shown by the expression of target genes of the DHH, PDGFRA, and NOTCH signaling. These observations indicate that NR2F2 is a permissive factor in steroidogenic progenitors acting downstream or in cooperation with the signaling pathways regulating FLC differentiation. NR2F2 expression has been found to be modulated by Hedgehog and/or NOTCH signaling in other cell types, and whether a similar regulation of *Nr2f2* expression by DHH and NOTCH signaling occurs in the steroidogenic progenitors remains unknown (*Lee et al., 2006*; *Swift et al., 2014*).

A fraction of FLC differentiates in *Nr2f2* mutants, including when progenitors of both coelomic epithelium and mesonephric origins are targeted. This observation indicates that additional factors cooperate with NR2F2 to regulate the transition from the steroidogenic progenitor state to the

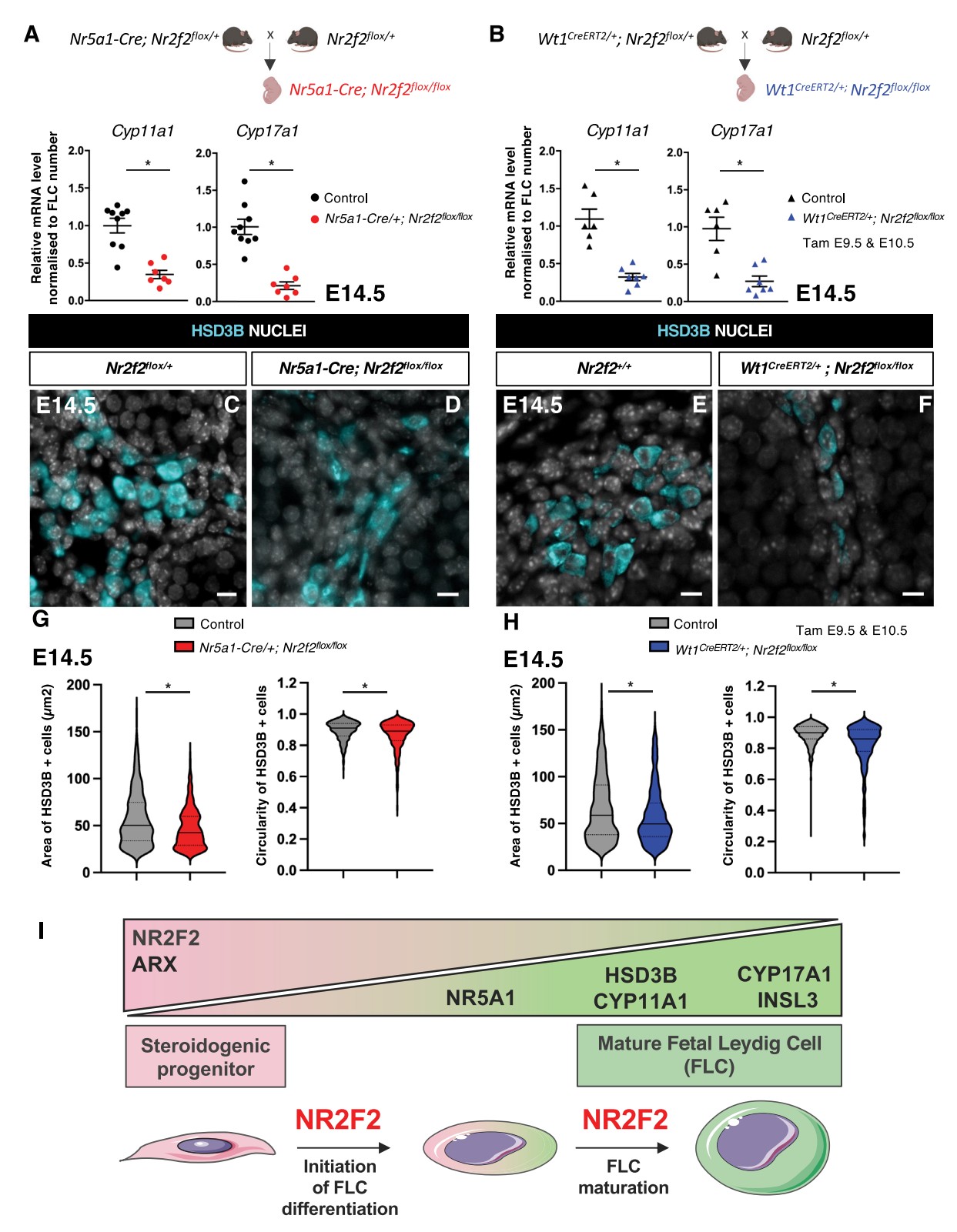

**Figure 5.** NR2F2 function is required for fetal Leydig cell (FLC) maturation. (**A**) RT-qPCR quantification of *Cyp11a1* and *Cyp17a1* transcripts in control (wild-type or *Nr2f2^flox/+^*) and *Nr5a1-Cre; Nr2f2^flox/flox^* testes after normalization to *Sdha* and *Tbp* and to the number of FLC as quantified by HSD3B immunofluorescence at embryonic day 14.5 (E14.5). Data are shown as means ± SEM. Statistical significance was assessed by Mann-Whitney U two-tailed test. * indicates p-value≤0.05; ns indicates p-value>0.05. (**B**) RT-qPCR quantification of *Cyp11a1* and *Cyp17a1* transcripts in *Nr2f2^flox/+^* and

*Figure 5 continued*

*Wt1^CreERT2; Nr2f2^flox/flox* gonads after normalization to *Sdha* and *Tbp* and to the number of FLC as quantified by HSD3B immunofluorescence at E14.5. Data are shown as means ± SEM. Statistical significance was assessed by Mann-Whitney U two-tailed test. * indicates p-value≤0.05; ns indicates p-value>0.05. (C,D) Immunodetection of HSD3B on E14.5 XY *Nr2f2^flox/+* and *Nr5a1-Cre; Nr2f2^flox/flox* testes. (E,F) Immunodetection of HSD3B on E14.5 XY *Nr2f2^flox/+* and *Wt1^CreERT2; Nr2f2^flox/flox* testes after tamoxifen was administered at E9.5 and E10.5. (G) Quantification of the area and circularity of HSD3B positive cells in two E14.5 control (wild-type or *Nr2f2^flox/+*, 737 cells, gray violin plot) and three *Nr5a1-Cre; Nr2f2^flox/flox* (486 cells, red violin plot) testes. Statistical significance was assessed by Mann-Whitney U two-tailed test. * indicates p-value≤0.05; ns indicates p-value>0.05. (H) Quantification of the area and circularity of HSD3B positive cells in three E14.5 control (wild-type or *Nr2f2^flox/+*, 1485 cells, gray violin plot) and three *Wt1^CreERT2; Nr2f2^flox/flox* (474 cells, blue violin plot) testes. Statistical significance was assessed by Mann-Whitney U two-tailed test. * indicates p-value≤0.05; ns indicates p-value>0.05. (I) Summary figure: NR2F2 is expressed in spindle-shaped interstitial steroidogenic progenitors together with ARX and is progressively downregulated upon FLC differentiation. NR2F2 is required for the initiation of FLC differentiation (marked by the upregulation of NR5A1) and for FLC maturation (characterized by the increase in cytoplasmic volume and the high expression of steroidogenic enzymes and *Insl3*). Immunodetection data are representative of triplicate biological replicates. Scale bar = 10 μm.

The online version of this article includes the following source data and figure supplement(s) for figure 5:

**Source data 1.** Source data for cell measurements and RT-qPCR data in *Figure 5*.

**Figure supplement 1.** NR2F2 function is required for fetal Leydig cell (FLC) maturation.

**Figure supplement 1—source data 1.** Source data for RT-qPCR data in *Figure 5—figure supplement 1*.

differentiating FLC. The homeodomain protein ARX is expressed in *Nr2f2* mutant steroidogenic progenitors and is a good candidate to be such a factor. *Arx* mutants exhibit decreased FLC numbers without defects in paracrine signals driving their differentiation, similar to *Nr2f2* mutants (*Miyabayashi et al., 2013*).

FLC and ALC are morphologically, transcriptionally, and functionally distinct (*Shima, 2019*; *Sararols et al., 2021*), yet NR2F2 function is required for the differentiation of both lineages. In contrast to the situation in the fetal testis, NR2F2 is maintained in cells that have started to express steroidogenesis genes in the postnatal testis (*Inoue et al., 2016*; *Ademi et al., 2022*; *Mendoza-Villarroel et al., 2014b*). NR2F2 regulates the transition from the adult progenitor Leydig cell (characterized by their elongated shape, their ability to proliferate, and a low level of testosterone synthesis) to the immature ALC (characterized by their round shape, low mitotic activity, and increased testosterone production) (*Qin et al., 2008*; *Haider, 2004*). In agreement with its in vivo role promoting maturation along the ALC lineage, NR2F2 cooperates with NR5A1 and GATA4 to activate the transcription of *Insl3*, *Star* (encoding the cholesterol transporter), and *Amhr2* (*Mendoza-Villarroel et al., 2014b*; *Mendoza-Villarroel et al., 2014a*; *Di-Luoffo et al., 2022*; *Mehanovic et al., 2021*; *Mehanovic et al., 2022*) in MA-10 cells.

Here, we found that the steroidogenic cells that differentiate in *Nr2f2* mutant fetal testes exhibit a small size, an elongated shape, and reduced steroidogenic gene expression, features of the earliest stages of FLC differentiation (*Wen et al., 2016*; *van den Driesche et al., 2012*; *Haider, 2004*). A similar phenotype is obtained when *Nr5a1* is deleted after the onset of FLC differentiation (*Buaas et al., 2012*). This suggests that in addition to controlling the initial engagement of steroidogenic progenitors into the FLC lineage and similar to the situation in the postnatal testis, NR2F2 promotes FLC maturation possibly by directly regulating the initial expression of genes involved in steroidogenesis. In agreement with this hypothesis, NR2F2 binding peaks are found in the regulatory regions of genes involved in lipid metabolism and cholesterol biosynthesis in ChIP-seq analysis of whole E14.5 testes (*Estermann et al., 2025*).

NR2F2 is required for additional aspects of fetal testis morphogenesis and differentiation. Testis cords are enlarged and abnormally shaped in *Nr2f2* mutants. Testis cord development involves the formation of Sertoli-germ cell masses after E11.5 and their subsequent partition by growing wedges of interstitial cells and associated vascular branches at E12.5 (*Cool et al., 2012*). How NR2F2-dependent regulation of interstitial cell adhesion or migration contributes to this process will be the aim of future research.

Pathogenic variants in NR2F2 have been associated with congenital malformations, including congenital heart disease, congenital diaphragmatic hernia, and syndromic 46,XX testicular or ovo-testicular difference/disorder in sex development (DSD) (*Polvani et al., 2019*; *Bashamboo et al., 2018*). More recently, defects in the external genitalia (micropenis, hypospadias) and cryptorchidism have been associated with rare heterozygous variants in NR2F2 in 46,XY patients (*Zidoune et al.,*

*2022*; *Ganapathi et al., 2023*; *Wankanit et al., 2024*). These phenotypes can be attributed to defects in testosterone-dependent masculinization and INSL3-dependent testis descent during gestation and could be explained by a failure of FLC differentiation in the fetal testis. NR2F2 is abundantly expressed in interstitial cells of fetal human testes, a population that likely contains the progenitors for FLC (*Kilcoyne et al., 2014*; *van den Driesche et al., 2012*; *Lottrup et al., 2014*; *Taelman et al., 2024*). The present work demonstrating that NR2F2 is required in the steroidogenic progenitors of the murine fetal testis for the initiation and progression of FLC differentiation provides an entry point in understanding the etiology of 46,XY DSD associated with pathogenic NR2F2 variants.

# Materials and methods

## Key resources table

| Reagent type (species) or resource | Designation | Source or reference | Identifiers | Additional information |
|---|---|---|---|---|
| Gene (*Mus musculus*) | *Nr2f2* | MGI | MGI:1352452 | |
| Genetic reagent (*Mus musculus*) | *Nr2f2tm1Vc* | Dr. M. Vasseur-Cognet | MGI:3578106 | |
| Genetic reagent (*Mus musculus*) | *Wt1tm2(cre/ERT2)Wtp* | Dr. William T Pu | MGI:7528785 | |
| Genetic reagent (*Mus musculus*) | *Tg(Nr5a1-cre)2Klp* | Dr. Keith L Parker | MGI:5493455 | |
| Chemical compound | Tamoxifen | Sigma-Aldrich | T5648 | 200 mg/kg body weight |
| Chemical compound | 5-Bromo-2'-deoxy-uridine | Sigma-Aldrich | B5002 | 50 mg/kg body weight |
| Chemical compound | Paraformaldehyde | EMS | 15710-S | 4% in PBS |
| Antibody | Anti-ACTA2 (Mouse monoclonal) | Gift from Dr. Chaponnier | | IF (1:500) |
| Antibody | Anti- Activated Caspase 3 (Rabbit polyclonal) | R&D Systems | AF835 (RRID:AB_2243952) | IF (1:200) |
| Antibody | Anti- AMH (Mouse monoclonal) | Bio-Rad | MCA2246 (RRID:AB_2226471) | IF (1:50) |
| Antibody | Anti- ARX (Rabbit polyclonal) | Gift from Pr. Morohashi and Dr. Inoue | | IF (1:200) |
| Antibody | Anti-COL4A1 (Rabbit polyclonal) | Abcam | ab19808 (RRID:AB_445160) | IF (1:400) |
| Antibody | Anti-CYP11A1 (Rabbit polyclonal) | Gift from Dr. Wilhelm | | IF (1:200) |
| Antibody | Anti-GATA4 (Goat polyclonal) | Santa Cruz Biotechnology | Sc-1237 (RRID:AB_2108747) | IF (1:200) |
| Antibody | Anti-GFP (Chicken polyclonal) | Abcam | Ab13970 (RRID:AB_300798) | IF (1:200) |
| Antibody | Anti-HSD3B (Goat polyclonal) | Santa Cruz Biotechnology | Sc-30820 (RRID:AB_2279878) | IF (1:200) |
| Antibody | Anti-HSD3B (Rabbit polyclonal) | Invitrogen | PA5-76669 (RRID:AB_2720396) | IF (1:500) |
| Antibody | Anti-Ki67 (Rabbit monoclonal) | Spring Bioscience | M3062 (RRID:AB_11219741) | IF (1:200) |
| Antibody | Anti-LAMA1 (Rabbit polyclonal) | Sigma-Aldrich | L9393 (RRID:AB_477163) | IF (1:200) |
| Antibody | Anti-NESTIN (Rabbit) | BioLegend | 839801 (RRID:AB_2565443) | IF (1:1000) |
| Antibody | Anti-NR2F2 (Mouse monoclonal) | R&D Systems | PP-H7147-00 (RRID:AB_2155627) | IF (1:200) |
| Antibody | Anti-NR5A1 (Rabbit polyclonal) | Cosmo Bio | KO611(RRID:AB_2861370) | IF (1:200) |
| Antibody | Anti-PDGFRA (Rabbit polyclonal) | Santa Cruz Biotechnology | SC-338 (RRID:AB_631064) | IF (1:200) |
| Antibody | Anti-PECAM-1 (Goat polyclonal) | Santa Cruz Biotechnology | Sc-1506 (RRID:AB_2161037) | IF (1:200) |
| Antibody | Anti-RUNX1 (Rabbit monoclonal) | Abcam | ab92336 (RRID:AB_2049267) | IF (1:500) |

*Continued on next page*

*Continued*

| Reagent type (species) or resource | Designation | Source or reference | Identifiers | Additional information |
|---|---|---|---|---|
| Antibody | Anti-SOX9 (Rabbit polyclonal) | Sigma-Aldrich | HPA001758 (RRID:AB_1080067) | IF (1:250) |
| Antibody | Anti-WT1 (Goat polyclonal) | R&D Systems | AF5729 (RRID:AB_2216239) | IF (1:200) |
| Commercial assay or kit | BrdU detection kit | Roche | 11 296 736 001 | |
| Commercial assay or kit | RNAscope Multiplex Fluorescent Reagent Kit v2 | Bio-Techne | 323110 | |
| Sequence-based reagent | *Gli1* probe | Bio-Techne | 311001 | |
| Commercial assay or kit | RNeasy Micro Kit | QIAGEN | 74004 | |
| Commercial assay or kit | SYBR Green I Master | Roche | 04887352001 | |
| Software, algorithm | RefFinder | https://www.ciidirsinaloa.com.mx/RefFinder-master/?type=reference | | |
| Software, algorithm | OMERO | https://www.openmicroscopy.org/omero/ | | |
| Software, algorithm | GraphPad Prism | Graphpad Software, Inc, La Jolla, CA, USA | GraphPad Prism 10.2.1 | |

## Mouse strains and genotyping

The experiments described herein were carried out in compliance with the guidelines of the French Regulations for Animal Care and with the approval of the local Ethical Committee (APAFIS APAF-IS#12789-2017121515109323 v1 and APAFIS#44072-2023061915491990 v5). Mouse lines were kept on a mixed background B6CBAF1/JRj. The *Nr2f2*$^{tm1Vc}$ line where *Nr2f2* exon 1 sequences (encoding the DNA binding domain) are deleted upon CRE-mediated recombination (referred to as *Nr2f2*$^{flox}$), the knock-in *Wt1*$^{tm2(cre/ERT2)Wtp}$ line where tamoxifen-inducible Cre$^{ERT2}$ is produced by WT1 expressing cells (referred to as *Wt1*$^{CreERT2}$), and the transgenic *Tg(Nr5a1-cre)2Klp* line where Cre expression is driven by *Nr5a1* regulatory sequences (referred to as *Nr5a1-Cre*) were genotyped as previously described (*Bardoux et al., 2005*; *Zhou et al., 2008*; *Bingham et al., 2006*). *Wt1*$^{CreERT2}$; *Nr2f2*$^{flox/+}$ or *Nr5a1-Cre*$^{tg/0}$; *Nr2f2*$^{flox/+}$ males were crossed with *Nr2f2*$^{flox/+}$ females to obtain mutant embryos at different stages. Embryos were named controls (*Nr2f2*$^{+/+}$ or *Nr2f2*$^{flox/+}$) or mutants (*Wt1*$^{CreERT2}$; *Nr2f2*$^{flox/flox}$ or *Nr5a1-Cre*$^{tg/0}$; *Nr2f2*$^{flox/flox}$). Genotypes of mice and embryos were determined using PCR assays on lysates from ear biopsies or tail tips. Genotyping primers are listed in Appendix 1. To activate the Cre$^{ERT2}$ recombinase in embryos, tamoxifen (TAM, T5648, Sigma-Aldrich) was directly diluted in corn oil to a concentration of 40 mg/mL, and two TAM treatments (200 mg/kg body weight) were administered to pregnant females by oral gavage at E9.5 and E10.5. For proliferation assays, 5-bromo-2'-deoxy-uridine (BrdU) (B5002, Sigma-Aldrich) diluted to a concentration of 10 mg/mL in sterile H$_2$O was administered to the pregnant females (50 mg/kg body weight) by intraperitoneal injection, and pregnant females and their embryos were humanely killed after 3 hr and 30 min. The day when a vaginal plug was found was designated as E0.5. E11.5–E12.5 embryos were staged by counting the number of ts with 18 ts corresponding to E11.5.

## Immunofluorescence staining and in situ hybridization

Embryos were fixed in 4% (wt/vol) paraformaldehyde (PFA, 15,710-S, EMS) overnight, processed for paraffin embedding, and sectioned into 5-µm-thick sections. Immunofluorescence and DAPI staining were performed as described in *Tang et al., 2020*. Proliferation analysis was performed by using a BrdU detection kit (11 296 736 001, Roche). *Gli1* mRNA was detected with the RNAscope technology (probe 311001) according to Advanced Cell's instructions using the RNAscope Multiplex Fluorescent Reagent Kit v2 Assay. Images were obtained on a motorized Axio Imager Z1 microscope (Zeiss) coupled with an AxioCam MRm camera (Zeiss) and processed with Fiji (Bethesda, MD, USA). The DAPI staining marking the nuclei was adjusted to visualize the tissues and may vary between samples.

However, for the immunofluorescence analysis, the exposure time of the acquisition of the fluorescent signal was identical in the same experiment to allow comparison between controls and mutants. Images were assembled using the open-source software platform OMERO (https://www.openmicroscopy.org/omero/). Antibodies are listed in the Key resources table. At least three embryos of each genotype were analyzed for each marker.

## Cell quantifications

The gonadal area for each section was measured by creating a gonadal region of interest (ROI) drawn manually in Fiji. The number of HSD3B positive cells (2–493 cells per section depending on the stages and genotypes), SOX9 positive cells (231–882 cells per section depending on the stages and genotypes), NR5A1 positive cells (3–256 cells per section depending on the stages and genotypes), ARX positive cells (666–2487 cells per section depending on the stages and genotypes), ARX positive cells that had incorporated BrdU (232–656 cells per section depending on the stages and genotypes), activated caspase 3 positive cells (0–6 cells per section depending on the stages and genotypes), or the number of nuclei labeled by DAPI (1582–4279 cells per section depending on the stages and genotypes) were counted manually in the entire gonadal section using the Cell Counter Plugin from Fiji. For each genotype, gonads of three or four biological replicates were analyzed. Two to three coronal sections spaced by at least 30 μm in the medial regions of the gonads were analyzed for each individual. Statistical significance was assessed by Mann-Whitney U two-tailed test. * indicates p-value≤0.05; ns indicates p-value>0.05.

## Quantification of area and circularity of HSD3B positive cells

Gonadal (ROI) was drawn manually, and HSD3B positive cells were segmented using the Stardist Deep Learning plugin of Fiji with a minimum area of 20 μm$^2$ to remove small particles. The area and circularity of each segmented cell were measured with Fiji. Circularity = 4π*area/perimeter^2. A value of 1.0 indicates a perfect circle. As the value approaches 0.0, it indicates an increasingly elongated shape. For each genotype, two to three biological replicates were analyzed. The data are shown as violin plots (with median and quartiles) for control and mutant samples.

## RNA extraction and quantitative PCR analysis

Individual gonads were dissected from the mesonephros in PBS, snap-frozen in liquid nitrogen, and kept at −80°C. RNA was extracted by RNeasy Micro Kit (74004, QIAGEN) and reverse-transcribed by M-MLV reverse transcriptase (M170A, Promega). The cDNA was used as a template for quantitative PCR analysis using the SYBR Green I Master (04887352001, Roche) and a LightCycler 480 System (Roche). Primer sequences are listed in Appendix 1.

All biological replicates of different genotypes (N=3–9) were run in the same plate and run as duplicate technical replicates. Relative gene expression of each gonad was normalized to the expression of the housekeeping genes *Shda* and *Tbp* (**Yokoyama et al., 2018**) by the 2-ΔΔCt calculation method. GeNorm, BestKeeper algorithms, and the comparative delta-Ct method provided through the online tool RefFinder (https://www.ciidirsinaloa.com.mx/RefFinder-master/?type=reference#) were used to confirm reference gene stability in the experimental datasets. Fold change in gene expression was obtained by dividing the normalized gene expression in gonads of a given genotype by the mean of the normalized gene expression in control gonads. Data are shown as means ± SEM. Statistical significance was assessed by Mann-Whitney U two-tailed test (GraphPad Prism 10.2.1). * indicates p-value≤0.05; ns indicates p-value>0.05.

## Acknowledgements

We acknowledge the help from members of the Experimental Histopathology Platform, the PRISM Imaging Platform, and the Animal House at iBV (Institut de Biologie Valrose, Université Côte d'Azur, CNRS, Inserm, iBV, France). We are grateful to members of the A Schedl, MC Chaboissier, and S Nef groups for helpful discussions and to D Wilhelm for critical reading of the manuscript. We are indebted to Dr. Chaponnier, Dr. Inoue, Pr Morohashi, Dr. M Vasseur-Cognet, and Dr. Wilhelm for sharing mouse lines and reagents. This research was funded by Agence Nationale de la Recherche ANR-23-CE14-0012, Heterosex, and by a scholarship from the China Scholarship Council (to FT).

# Additional information

## Funding

| Funder | Grant reference number | Author |
|---|---|---|
| Agence Nationale de la Recherche | ANR-23-CE14-0012 | Marie-Christine Chaboissier |
| China Scholarship Council | | Furong Tang |
| Agence Nationale de la Recherche | Heterosex | Marie-Christine Chaboissier |

The funders had no role in study design, data collection and interpretation, or the decision to submit the work for publication.

## Author contributions

Aitana Perea-Gomez, Conceptualization, Formal analysis, Supervision, Investigation, Methodology, Writing - original draft, Writing - review and editing; Natividad Bellido Carreras, Magali Dhellemmes, Furong Tang, Coralie Le Gallo, Investigation; Marie-Christine Chaboissier, Conceptualization, Formal analysis, Supervision, Funding acquisition, Writing - original draft, Project administration, Writing - review and editing

## Author ORCIDs

Aitana Perea-Gomez ⓘ https://orcid.org/0000-0002-7788-038X
Natividad Bellido Carreras ⓘ http://orcid.org/0000-0003-1207-0711
Magali Dhellemmes ⓘ https://orcid.org/0000-0001-7983-3221
Furong Tang ⓘ http://orcid.org/0000-0001-7546-7500
Coralie Le Gallo ⓘ http://orcid.org/0009-0000-5079-4791
Marie-Christine Chaboissier ⓘ https://orcid.org/0000-0003-0934-8217

## Ethics

The experiments described herein were carried out in compliance with the relevant institutional and French animal welfare laws, guidelines, and policies (APAFIS APAFIS#12789-2017121515109323 v1 and APAFIS#44072-2023061915491990 v5).

Reviewer #1 (Public review): https://doi.org/10.7554/eLife.103783.3.sa1
Reviewer #2 (Public review): https://doi.org/10.7554/eLife.103783.3.sa2
Author response https://doi.org/10.7554/eLife.103783.3.sa3

# Additional files

## Supplementary files

MDAR checklist

## Data availability

All data generated or analysed during this study are included in the manuscript and supporting files.

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

# Appendix 1

## Primers

| Primer name | Application | Sequence (5'-3') |
| --- | --- | --- |
| CRE 1 | Genotyping *Cre* | CAGGATATACGTAATCTGGC |
| CRE 4 | Genotyping *Cre* | CACGGGCACTGTGTCCAGACCAG |
| CCR5mR | Genotyping internal control | ATGTGGATGGAGAGGAGTCG |
| CCR5mL | Genotyping internal control | CAACCGAGACCTTCCTGTTC |
| TF2.1 | Genotyping *Nr2f2* | TGCCCACACTTTCCTACTCC |
| TF2.5 | Genotyping *Nr2f2* | TTTCTGCAAGGAATGGGTTG |
| *Amh-Fwd* | RT-qPCR primers | GGGGAGACTGGAGAACAGC |
| *Amh-Rev* | RT-qPCR primers | AGAGCTCGGGCTCCCATA |
| *Arx-Fwd* | RT-qPCR primers | GCACCACGTTCACCAGTTAC |
| *Arx-Rev* | RT-qPCR primers | GCACCACGTTCACCAGTTAC |
| *Bhmt-Fwd* | RT-qPCR primers | CGTCAGCTTCATCGGGAGTT |
| *Bhmt-Rev* | RT-qPCR primers | CTTGCCGTGCAATGTCACAA |
| *Cdh5-Fwd* | RT-qPCR primers | TCCTCTGCATCCTCACCATCACA |
| *Cdh5-Rev* | RT-qPCR primers | GTAAGTGACCAACTGCTCGTGAAT |
| *Cyp11a1-Fwd* | RT-qPCR primers | TGGCCCCATTTACAGGGAGAA |
| *Cyp11a1-Rev* | RT-qPCR primers | GGCATCTGAACTCTTAAACAGGA |
| *Cyp17a1-Rev* | RT-qPCR primers | CAGAGAAGTGCTCGTGAAGAAG |
| *Cyp17a1Fwd* | RT-qPCR primers | CAGAGAAGTGCTCGTGAAGAAG |
| *Dhh-Fwd* | RT-qPCR primers | GGACCTCGTACCCAACTACAA |
| *Dhh-Rev* | RT-qPCR primers | CGATGGCTAGAGCGTTCACC |
| *Gli1-Fwd* | RT-qPCR primers | TGGTACCATGAGCCCTTCTT |
| *Gli1-Rev* | RT-qPCR primers | GTGGTACACAGGGCTGGACT |
| *Hes1-Fwd* | RT-qPCR primers | ATAGCTCCCGGCATTCCAAG |
| *Hes1-Rev* | RT-qPCR primers | ATAGCTCCCGGCATTCCAAG |
| *HeyL-Fwd* | RT-qPCR primers | CAGCCCTTCGCAGATGCAA |
| *HeyL-Rev* | RT-qPCR primers | CCAATCGTCGCAATTCAGAAAG |
| *Insl3-Fwd* | RT-qPCR primers | ATTGCTCCCCACCTCCTGGCTATG |
| *Insl3-Rev* | RT-qPCR primers | GGTCATGATGGGGCTTCTTGGGGA |
| *Notch2-Fwd* | RT-qPCR primers | ATATCGACGACTGCCCCAAC |
| *Notch2-Rev* | RT-qPCR primers | CCATAGCCTCCGTTTCGGTT |
| *Nr2f2-Fwd* | RT-qPCR primers | CGGAGGAACCTGAGCTACAC |
| *Nr2f2-Rev* | RT-qPCR primers | CGGAGGAACCTGAGCTACAC |
| *Pdgfa-Fwd* | RT-qPCR primers | CAGTGTCAAGGTGGCCAAAGT |
| *Pdgfa-Rev* | RT-qPCR primers | CAGTGTCAAGGTGGCCAAAGT |
| *Pdgfb-Fwd* | RT-qPCR primers | GATCTCTCGGAACCTCATCG |
| *Pdgfb-Rev* | RT-qPCR primers | GATCTCTCGGAACCTCATCG |

*Continued on next page*

*Continued*

| Primer name | Application | Sequence (5'-3') |
| --- | --- | --- |
| *Pdgfra-Fwd* | RT-qPCR primers | TCCATGCTAGACTCAGAAGTCA |
| *Pdgfra-Rev* | RT-qPCR primers | TCCCGGTGGACACAATTTTT |
| *Sdha-Fwd* | RT-qPCR primers | TGTTCAGTTCCACCCCACA |
| *Sdha-Rev* | RT-qPCR primers | TCTCCACGACACCCTTCTG |
| *Sgpl1-Fwd* | RT-qPCR primers | CTGAAGGACTTCGAGCCTTATTT |
| *Sgpl1-Rev* | RT-qPCR primers | CTGAAGGACTTCGAGCCTTATTT |
| *Sox9-Fwd* | RT-qPCR primers | GCGGAGCTCAGCAAGACTCTG |
| *Sox9-Rev* | RT-qPCR primers | ATCGGGGTGGTCTTTCTTGTG |
| *Tbp-Fwd* | RT-qPCR primers | GCTCTGGAATTGTACCGCAG |
| *Tbp-Rev* | RT-qPCR primers | TGACTGCAGCAAATCGCTTG |

