## [Editor Report · eLife Assessment]

This **important** study, which has been improved further upon revision, reveals a critical role of the transcription factor NR2F2 in mouse fetal Leydig cell (FLC) differentiation. With elegantly carried out experiments, the authors provide **compelling** evidence that NR2F2 helps to initiate the differentiation of certain interstitial cells into FLC until these cells mature into functional secretory cells that produce androgen and insulin-like peptide 3 (INSL3). The particular importance of the work comes from the fact that NR2F2 affects FLCs without altering paracrine signals known to be involved in FLC differentiation. The work will be of interest to colleagues studying reproductive development in mammals including humans or the biological functions of the nuclear receptor family.

---

## [Referee Report · Reviewer #1 (Public review)]

Summary:

In this beautiful paper the authors examined the role and function of NR2F2 in testis development and more specifically on fetal Leydig cells development. It is well known by now that FLC are developed from an interstitial steroidogenic progenitor at around E12.5 and are crucial for testosterone and INSL3 production during embryonic development, which in turn shapes the internal and external genitalia of the male. Indeed, lack of testosterone or INSL3 are known to cause DSD as well as undescended testis, also termed as cryptorchidism.

The authors first characterized the expression pattern of the NR2R2 protein during testis development and then used two cKO systems of NR2F2, namely the Wt1-creERT2 and the Nr5a1-cre to explore the phenotype of loss of NR2F2. They found in both cases that mice are presenting with undescended testis and major reduction in FLC numbers. They show that NR2F2 has no effect on the amount and expression of the progenitor cells but in its absence, there are less FLC and they are immature.

The effect of NR2F2 is cell autonomous and does not seem to affect other signalling pathways implemented in Leydig cell development as the DHH, PDGFRA and the NOTCH pathway.

Overall, this paper is excellent, very well written, fluent and clear. The data is well presented, and all the controls and statistics are in place. I think this paper will be of great interest to the field and paves the way for several interesting follow up studies as stated in the discussion

Comments on revised version:

The authors have fully addressed my concerns and the manuscript is looking excellent.

---

## [Referee Report · Reviewer #2 (Public review)]

The major conclusion of the manuscript is expressed in the title: "NR2F2 is required in the embryonic testis for Fetal Leydig Cell development" and also at the end of the introduction and all along the result part. All the authors' assertions are supported by very clear and statistically validated results from ISH, IHC, precise cell counting and gene expression levels by qPCR. The authors used two different conditional Nr2f2 gene ablation systems that demonstrate the same effects at the FLC level. They also showed that the haplo-insufficiency of Wt1 in the first system (knock-in Wt1-cre-ERT2) aggravated the situation in FLC differentiation by disturbing the differentiation of Sertoli cells and their secretion of pro-FLC factors, which had a confounding effect and encouraged them to use the second system. This demonstrates the great rigor with which the authors interpreted the results. In conclusion, all authors' claims and conclusions are justified by their high-quality results.

Comments on revised version:

In their revised version, the authors have taken full account of all my suggestions, and I congratulate them on this. I have no further comments to make on this new version.

---

## [Author Response]

The following is the authors’ response to the original reviews

**Public Reviews:**

**Reviewer #1 (Public review):**
SummaryIn this beautiful paper the authors examined the role and function of NR2F2 in testis development and more specifically on fetal Leydig cells development. It is well known by now that FLC are developed from an interstitial steroidogenic progenitors at around E12.5 and are crucial for testosterone and INSL3 production during embryonic development, which in turn shapes the internal and external genitalia of the male. Indeed, lack of testosterone or INSL3 are known to cause DSD as well as undescended testis, also termed as cryptorchidism. The authors first characterized the expression pattern of the NR2R2 protein during testis development and then used two cKO systems of NR2F2, namely the Wt1-creERT2 and the Nr5a1-cre to explore the phenotype of loss of NR2F2. They found in both cases that mice are presenting with undescended testis and major reduction in FLC numbers. They show that NR2F2 has no effect on the amount and expression of the progenitor cells but in its absence, there are less FLC and they are immature.The effect of NR2F2 is cell autonomous and does not seem to affect other signalling pathways implemented in Leydig cell development as the DHH, PDGFRA and the NOTCH pathway.Overall, this paper is excellent, very well written, fluent and clear. The data is well presented, and all the controls and statistics are in place. I think this paper will be of great interest to the field and paves the way for several interesting follow up studies as stated in the discussion
**Reviewer #2 (Public review):**
The major conclusion of the manuscript is expressed in the title: "NR2F2 is required in the embryonic testis for Fetal Leydig Cell development" and also at the end of the introduction and all along the result part. All the authors' assertions are supported by very clear and statistically validated results from ISH, IHC, precise cell counting and gene expression levels by qPCR. The authors used two different conditional Nr2f2 gene ablation systems that demonstrate the same effects at the FLC level. They also showed that the haplo-insufficiency of Wt1 in the first system (knock-in Wt1-cre-ERT2) aggravated the situation in FLC differentiation by disturbing the differentiation of Sertoli cells and their secretion of pro-FLC factors, which had a confounding effect and encouraged them to use the second system. This demonstrates the great rigor with which the authors interpreted the results. In conclusion, all authors' claims and conclusions are justified by their high-quality results.
**Recommendations for the authors:**

We thank the reviewers for their comments which have improved and strengthened our manuscript. Please see our responses to specific comments below in blue.

**Reviewer #1 (Recommendations for the authors):**
I have several small comments:(1) There has been recently a preprint from the Yao lab about the role of NR2F2 is steroidogenic cells (https://www.biorxiv.org/content/10.1101/2024.09.16.613312v1). They performed cKO of NR2F2 using the Wt1creERT2 and found similar results. You should present and discuss this paper in light of your results.

Estermann et al., report a very similar phenotype of FLC hypoplasia in an independent mouse model of *Nr2f2* conditional mutation. We have now referred to this article in the discussion of our manuscript as suggested.

(2) In the introduction I think it is important to mention that the steroidogenic progenitors are derived from Wnt5a positive cells (https://pubmed.ncbi.nlm.nih.gov/35705036/).

We have mentioned this point in the introduction as suggested.

(3) In both models you show a decrease in the number of FLC (60% or 40%) and yet they both present with undescended testis. It is important to discuss the fact that there is no need for a complete ablation of testosterone and INSL3 in order to get cryptorchidism.

We have mentioned this point in the discussion as suggested.

The fact that you get only partial reduction in FLC is likely due to redundancy with additional factors, possibly the ARX like you stated in the discussion and it will be interesting to explore that in the future but is beyond the scope of the current paper.

We agree with the reviewer, this question could be addressed by analyzing *Arx,Nr2f2* double mutants.

(4) In page 8 line 11 you mention data not shown- not sure if this is allowed in the journal .

The data is now shown in Figure S5A as suggested.

(5) In Figure 2- it will be good if you add a schematic model of the mouse strains used as well as the experimental and control mice next to the Tam scheme. Similar scheme should be in figure 3 for Nr5a1-cre.

We have modified Figures 2 and 3 as suggested.

(6) There is a clear and pronounced effect of the testis cords number and size. It will be good if you could qualify testis cord numbers/ diameter in the mutants even if you do not follow in detail the effect on Sertoli cells

We have quantified testis cords numbers and area in E14.5 Control and Wt1^CreERT2/+^; *Nr2f2flox/flox* testes. This data is now shown in Figure S2M.

(7) It will be good to present the undescended testis in the Wt1-cre model in figure 2 and not in the supp figure

The data is now shown in Figure 2H-I as suggested.

(8) Please add labelling of the testis, kidney, bladder, vas deferens in figure 3 N+O and in the Wt1-cre model

We have added the labels in Figures 2 and 3 as suggested.

(9) In figure 5 which present both models- it will be good to use the scheme I suggested before to highlight which results refer to which ko model.

We have modified Figure 5 as suggested.

**Reviewer #2 (Recommendations for the authors):**
The work presented in this manuscript gave me food for thought. I have always been intrigued by the fact that of the large number of interstitial cells in the testis, a minority differentiate into mature androgen-producing Leydig cells. In other words, how is the number of functional steroidogenic cells defined from a large pool of progenitor cells (ARX and NR2F2 positive ones)? This may have a link with the levels of androgens produced (a kind of feedback control) or the effectiveness of these androgens on the target tissues (i.e.: as spermatogenesis efficiency in adults). In addition, there must be specific signals (probably linked to gonadotropins) that induce the recruitment of Leydig cells from the progenitor pool. Perhaps the genetic models generated in this study could help to address these questions. I leave it to the authors to judge.

We agree with the reviewer. How NR2F2 (and other factors) integrate extrinsic cues to regulate the recruitment of a subset of interstitial steroidogenic progenitors along the Leydig cell differentiation pathway is a fascinating question beyond the scope of this work.

In addition to this reflection, I propose a few minor modifications likely to improve the quality of the manuscript:(1) Page 3, lane 3: I suggest to replace "growth" by "differentiation"

We have modified the text as suggested.

(2) Page 3, lane 4: the "scrotum" is missing in the parenthesis. Please add it before "and penis"

We have modified the text as suggested.

(3) Page 5, lanes 21-24: kidney hypoplasia is also evident on Fig S2H (stated in the figure legend). It could be also mentioned in this sentence and it implies "...that NR2F2 function is required for testicular and kidney development."

We have modified the text as suggested.

(4) Page 5, lanes 28-30. In addition to the reduction in the number of HSD3B-positive cells, HSD3B staining seems clearly more faint in mutant FLC (Fig 2M) compared to adrenal cells on the same section or FLC in control gonads. This fits well with other results on the level of steroidogenic enzymes (Fig 2O) and those presented thereafter (Fig S4 I-J and Fig 5). Perhaps the author could mention this fact.

We have modified the text as suggested in the results section “NR2F2 is required for FLC maturation” (Page 8).

(5) Page 5, lanes 31-34: testicular descent is hugely sensible to INSL3 in the mouse (by contrast with other species where androgens seem to be more critical). I was wondering if you can check a better phenotypic marker for the absence (or reduction) of androgens like the differentiation of epididymides by HE staining or the anogenital distance at birth.

We have measured the anogenital distance at P0 and P1 as suggested and have included the corresponding graph in Fig. S3P

(6) Page 8, lanes 21-22: "HSD3B positive FLC were smaller and more elongated". It is clear on Fig 5F but not evident on Fig 5D. Could the authors propose another image?

We have modified Figure 5 as suggested and provide now another example of HSD3B positive FLCs in a *Nr5a1Cre; Nr2f2flox/flox* mutant gonad (Fig. 5D) and the corresponding control littermate (Fig. 5C).

(7) Page 14, lane 12: "(arrow in I)" should be "(arrow in H)"

We have modified the text as suggested. Please note that ACTA 2 expression is now shown in Figure S2 G-H.

(8) Page 15, lane 6: "Arrows indicate NR5A1 positive FLC". There is no arrow on Fig4 C,D; but a kind of scale bar on the enlargement shown in C.

We have modified Figure 4 as suggested.